# Awareness of Locomotive Syndrome and Factors Associated with Awareness: A Community-Based Cross-Sectional Study

**DOI:** 10.3390/ijerph17197272

**Published:** 2020-10-05

**Authors:** Keiko Sugai, Haruhiko Imamura, Takehiro Michikawa, Keiko Asakura, Yuji Nishiwaki

**Affiliations:** 1Department of Orthopaedic Surgery, Keio University School of Medicine, 35 Shinanomachi, Shinjuku, Tokyo 160-8582, Japan; ksugai@keio.jp; 2Department of Environmental and Occupational Health, School of Medicine, Toho University, 5-21-16 Omorinishi, Ota-ku, Tokyo 143-8540, Japan; takehiro.michikawa@med.toho-u.ac.jp (T.M.); keiko.asakura@med.toho-u.ac.jp (K.A.); yuuji.nishiwaki@med.toho-u.ac.jp (Y.N.)

**Keywords:** locomotive syndrome, elderly health, cross-sectional study, community based study, preventive care

## Abstract

Locomotive syndrome is a condition of reduced mobility, and patients have a high risk of requiring nursing care. In order to investigate the level of awareness of the term “locomotive syndrome” and the factors relating to awareness in a community, awareness of locomotive syndrome was included in a questionnaire survey on health and daily life conducted in Koumi Town (Japan), which was distributed to 3181 eligible residents aged 40 years or older. Information on age, sex, marital status, educational attainment, lifestyle, and social environment was also collected, and the association of awareness with various factors was analyzed with two multivariable Poisson regression models. As a result, awareness among respondents was 44.6%. Awareness was significantly higher among women, those who were 60–79 years old, married, and had received higher education. Additionally, awareness was significantly associated with social factors, especially attendance at regional events within the last one year, in both women and men: the adjusted prevalence ratios (95% confidence intervals) were 1.26 (1.10–1.43) and 1.48 (1.19–1.83), respectively. In conclusion, in addition to strengthen awareness rising campaigns targeting men and for younger people, providing health education at social settings such as regional events may help improve future musculoskeletal health in the elderly.

## 1. Introduction

Population ageing is now a global phenomenon. The proportion of population aged 65 years or over increased from 6% in 1990 to 9% in 2019, and is projected to rise to 16% by 2050 [1]. Japanese society is said to be aging at the fastest rate in the world; the proportion of people aged 65 years or over as of 2019 is 28.5% [2], which is the highest proportion in the world. Ageing of society accompanies a risk to increase financial pressure on old-age support systems. Actually in Japan, there is a gap of 9 to 12 years between healthy life expectancy and average life expectancy, which places a heavy burden on nursing care [3]. Musculoskeletal problems (fractures and articular diseases) are the most common cause of the need for nursing care [4], so the importance of maintaining musculoskeletal health is gathering attention in preventing the care-need.

Locomotive syndrome is a term coined in 2007 by the Japanese Orthopaedic Association [5,6,7,8,9]. It involves reduced mobility due to diminished musculoskeletal performance, and patients have a high risk of requiring future nursing care. The progression of locomotive syndrome can be assessed by test composed of two physical exams and one self-reported questionnaire [10]. Using this test, about 81.0% and 34.1% of Japanese people at the age of 60 or over are reported be at the 1st (the beginning of decline in mobility) and the 2nd (progressing towards decline in mobility) stage of locomotive syndrome [11], respectively. The prevalence increases with age [12], because the conditions underlying it are more common in older people [6,12]. For example, lumbar spondylosis (Kellgren-Lawrence > 2) is present in 81.5% of men and 65.5% of women over 40 years old; the corresponding figures for radiographic knee osteoarthritis are 42.6% and 62.4% [13]. In addition to aging, lifestyle factors such as lack of exercise and inadequate nutritional intake also cause progression of locomotive syndrome. Excessive loading and obesity are known to be risk factors for the progression of deformation and impairment of joints and intervertebral disks, and insufficient loading and extreme thinness are risk factors for osteoporosis and sarcopenia [6]. Because the symptoms of the conditions underlying locomotive syndrome appear only after the condition has progressed to some extent, it may be too late to start taking preventive measures after symptoms appear. To effectively prevent physical decline, therefore, preventive measures need to be initiated before the condition progresses, which cannot be achieved without raising public awareness of locomotive syndrome [14].

Many studies have shown the effectiveness of physical interventions in preventing physical decline in elderly people [15,16,17]; although, their effects are limited for severe disabilities [16]. This points up the importance of implementing preventive measures before the onset of symptoms, or while they are still mild; however, knowledge among the public of the risks of locomotive syndrome is necessary to achieve this. There are many reports that awareness of disease leads to preventive behavior [18,19,20]. Likewise, awareness of locomotive syndrome is necessary for people to take preventive measure in their daily life.

The Japanese Ministry of Health, Labour, and Welfare aims has set a target of increasing the percentage of individuals who know about locomotive syndrome to 80% by 2022 from that of 17.3% in 2012 through its national health promotion program “The second term of National Health Promotion in the twenty first century (Health Japan 21 (the second term))” (2013–2022) [6,21]. Awareness-raising activities have succeeded to some extent [7,22], but the level of public awareness of the syndrome has leveled off at around 45% since 2015 [23,24]. Although it is important to raise awareness of the syndrome to promote preventive action, it is difficult to find effective publicity measures.

In this study, we investigated the level of awareness of locomotive syndrome and the factors associated with awareness in a rural community. We hope that the introduction of our initiatives of super-aging Japanese society will serve as a reference for ageing measures in other countries.

## 2. Materials and Methods

### 2.1. Study Population

This study was performed as part of a questionnaire survey on health and daily life conducted in Koumi Town, Nagano Prefecture. Koumi Town is a mountainous community about 100 km northwest of Tokyo, Japan, with a population of approximately 4700 and relatively high proportion of people aged 65 and over (39.1% in 2015). In August 2017, targeting all residents aged 40 years or older who were not certified for nursing care, the questionnaire was distributed to 3316 residents with the support of Koumi Town Hall and health promotion volunteers. Excluding those who were hospitalized or institutionalized, 3181 residents were approached, identified as eligible for participation in the study, and received the questionnaire. By anonymous posting, 1893 of them (59.5%) returned the questionnaire. Of these respondents, 1804 (56.7%) answered the question about their awareness of locomotive syndrome. In this sample size (*n* = 1804), the awareness proportion of 45% [23,24] could estimate 95% confidence interval with less than 5%. The schema of the recruitment of the participants are shown as Figure 1.

The study protocol was approved by the institutional ethics committee of the Faculty of Medicine, Toho University, Tokyo (Approval No. A19068_A17095_A17048). All participants were informed of the purpose and methods of the study, their right to participate or not, and their right to withdraw; they were also assured of anonymity.

### 2.2. Awareness of Locomotive Syndrome

Awareness was assessed with the following question: “Do you know the term locomotive syndrome?” The response options were “No,” “I have heard the term but do not know what it means,” and “I have heard the term and know what it means.” For the purposes of this study, the responses were sorted into two categories: “No” for the first option, and “Yes” for the other two. This definition of awareness is the same as in Health Japan 21 [21].

### 2.3. Participant Characteristics

Information on sex and age was collected from the town registry. Information was collected from the questionnaire responses on marital status (single, divorced, or widowed/married) and educational attainment (junior high or lower/high school or higher), and also on lifestyle habits: current alcohol drinking habit (no/yes), current smoking habit (no/yes), and weekly exercise habit (less than one hour/one hour or more). To assess the association of awareness of locomotive syndrome with social factors, the participants were asked about their participation in any of the group activities such as local activities (membership of neighborhood associations, women’s associations, senior citizens’ associations, etc.), sports, hobbies, entertainment activities, volunteer activities, nonprofit organization/citizen activities, voluntary financial cooperative associations, group associations, and other activities (membership of business associations, industry associations, etc.), and responses on all of the above were classified according to the frequency of participation (several times a year in at least one/none). Attendance at any of the following regional events within the previous one year was also recorded (yes, attended at least one/no, or could not remember): health promotion/disease prevention classes, cultural festivals or health welfare festivals, and social education classes. In order to assess the extent of the participants’ social interaction, they were asked if they had someone they could talk with about their worries and complaints, about their own health problems, or someone who took care of them when they were sick; they were also asked if they had someone whose worries and complaints they listened to, or someone they took care of when he/she was sick. Responses were categorized as yes for at least one, or no. Relations with neighbors were also assessed: daily contact and cooperation in neighborhood chores, or minimal or no contact. From the women, we obtained information about whether they had any experience of health promotion volunteers (none/yes). In Koumi Town, about 65 women have been recruited to work as health promotion volunteers under the supervision of a municipal public health nurse every two years since 1975. They support health promotion activities for families in the local community by, for example, giving recommendations for health examinations.

### 2.4. Statistical Analysis

We used chi-square tests for proportions. Furthermore, the participants were divided into two groups according to their awareness of locomotive syndrome, sex separately, and the baseline variables between the two groups were compared.

The association of awareness of locomotive syndrome with various factors was analyzed sex-separately with two multivariable Poisson regression models, considering the disparities in awareness and background factors between men and women. Model 1 included the basic characteristics of age, marital-status, and educational attainment. Model 2 also included alcohol drinking habit, smoking habit, weekly exercise habit, participation in group activities, attendance at regional events within one year, social support, contact with neighbors, and experience of health promotion volunteering (women only). The strengths of the associations were represented by prevalence ratios (PRs) and 95% confidence intervals (95% CIs).

Stata version 14 (Stata Corp., College Station, TX, USA) was used for all analyses.

## 3. Results

### 3.1. Participant Characteristics and Awarenss of Locomotive Syndrome

Significantly more of the participants (*n* = 1804) were in their 60s and 70s than the non-participants (*n* = 1377), and women also represented a significantly larger proportion of the participants (Appendix A). Of the participants, 437 (24.2%) had heard the term locomotive syndrome but did not know its meaning, and 367 (20.3%) had heard the term and also understood it, meaning that the total awareness proportion was 44.6% (804). The proportion differed significantly by sex: 32.2% of the men, compared with 54.4% of the women (*p* < 0.001). The distributions of age, marital status, and educational attainment were similar in both sexes, but participants who were 60–79 years old, married, and had higher educational attainment showed greater awareness of locomotive syndrome; the women with experience of health promotion volunteers had a significantly higher awareness proportion (Table 1).

### 3.2. Prevalence Ratios of Each Covariates to Increase Awarenss of Locomotive Syndrome

Table 2 and Table 3 summarize the adjusted PRs for each factor affecting awareness according to sex. Attendance at regional events within the last one year was significantly associated with awareness of locomotive syndrome in both women and men, with adjusted PRs (95% CI) in Model 2 of 1.26 (1.10–1.43) and 1.48 (1.19–1.83), respectively. Participation in group activities tended to increase awareness in both sexes. Higher educational attainment was well associated with awareness among the women, and weakly associated among the men. Closer relations with neighbors only showed an association among the women. Additionally, experience of health promotion volunteering (only applicable to women) was significantly associated with awareness (PR (95% CI) 1.25 (1.09–1.43) in Model 2) (Table 2). On the other hand, lifestyle factors (drinking alcohol, smoking, exercising) were found to have no association with awareness.

## 4. Discussion

The public awareness of locomotive syndrome in this study was consistent with those of a nationwide internet survey carried out in 2020 [24], showing that awareness is comparatively high among people in their 60s and 70s, when the symptoms of locomotive syndrome become apparent [10,11,12,25]. The age-related difference of awareness remained after the multivariate adjustment, indicating that the awareness-rising promotion measures need to be focused more on younger people. For these people, educational activities in the workplace may be more effective in increasing awareness than encouraging them to attend regional events. Intervention for this age group is especially important, in order to take preventive measures early enough. On the other hand, the awareness rising activity need to be conducted for elderly people as well, since it has been reported that exercise is effective to some extent to reduce the risk of requiring nursing care even for those who have already developed locomotive syndrome [26]. Home visits and specific guidance targeted at the elderly during regional events may be a good way to increase awareness in this age group.

The difference in awareness between the sexes was more pronounced in our study than in the 2019 nationwide internet survey: we found that 32.2% of the men and 54.4% of the women were aware of the term, as compared with 41% of the men and 48.6% of the women in the internet survey [23]. The reason for the higher awareness proportion among our female participants may be that 44.4% of those aged over 40 had experience of health promotion volunteers in Koumi Town. Health promotion volunteering is a traditional activity in Japan, in which volunteers operate under the supervision of a municipal public health nurse [27]. In the Koumi area, only women have been assigned historically. They engage in various activities, such as giving advice on nutrition and promoting blood pressure checks in their community, which is effective in educating them about health issues. Attempts to make people take responsibility for others health in this way may be useful, for men, as well.

It was a novel information that social factors such as attendance at regional health-related events were associated with awareness. Closer relations with neighbors were also significantly associated with awareness among the female participants, indicating that communication with others may play an important role in improving health-related knowledge. However, the direction of causal relationship could not be elucidated from this research. That is, sociable people may have acquired health-related knowledge through regional events and conversations in the neighborhood. In this case, promotion of educational activities at regional events may work to further improve the awareness. On the other hand, there is also a possibility that people with knowledge of locomotive syndrome may be more active in regional health-related events as a result of being aware of importance of active lifestyles; it is a good sign that the recognition lead to the change of life-style. In the latter case, presenting some specific exercise using the place of the regional events may contribute to further improve the musculoskeletal performance of participants.

Looking back, in Japan, awareness of metabolic syndrome has dramatically increased since it was taken up as a basic medical examination item targeting middle-aged and older adult in 2008 [28,29]. According to this, incorporating locomotive syndrome into the health checkup items may be the most effective way to raise awareness. Already, the health-survey for musculoskeletal system has been added to the heath-examination items targeting school children since 2016. In the future, similar approach targeting adults could be useful in raising interest in the musculoskeletal health in younger adults as well as older adults. However, there may still be a hurdle for government to adopt the evaluation of musculoskeletal function as a public health checkup item because locomotive syndrome is not a disease that requires immediate treatment. Moreover, the improvement of the locomotive function may rely largely on individual efforts. Therefore, considering the current social situation, Japan is focusing on raising awareness of the locomotive syndrome, as the first stage to cope with the condition. We believe that our research results are useful as one of the keys to further increase awareness.

Considering the actions to improve the mobility in people, exercises composed of single-leg standing and squatting are recommended from the Japanese Orthopaedic Association and is named Locomotion Training (Locotra) [30]. These exercises aim to strengthen standing and gait functions and can be safely performed at home [6]. Additional exercises, stretches, as well as balanced diet that helps strengthen bones and muscles are also recommended by the Japanese Orthopaedic Association [14,30], and various other approaches has been reported as well [31,32]. Though it has not yet been confirmed that musculoskeletal degeneration can be treated by exercise, promoting physical activity appears promising.

The main strength of this study compared to the annual nationwide internet survey [23,24] is that ours is the first community-based cross-sectional study of public awareness of locomotive syndrome. Although the previous internet surveys [23,24] targeted 10,000 men and women over 20 years old, and employed a sampling system that approximately matched participants across prefectures with national census data in terms of population, sex, and age, questions remain about the generalizability of its results: it may, for example, have targeted a smaller proportion of elderly people than exist in the general population, and a larger proportion of technologically savvy people with internet access. Furthermore, internet surveys have data integrity concerns, such as low participation proportions, false answers, and multiple replies from the same respondent [33]. In comparison, our study covered approximately 60% of the residents aged 40 years or older of a specific area and may well have had a smaller sampling bias than the internet survey. Our research also evaluated the association of awareness of locomotive syndrome with social and lifestyle factors for the first time, providing useful hints on effective ways to increase awareness in the future.

On the other hand, our study has four limitations. First, it might still have been subject to selection bias, because the age and sex of the participants did not match those of the non-participants. There is also a possibility that the survey attracted a disproportionally large number of people who are particularly interested in health issues. Secondly, since the study was conducted in one specific town, it may be difficult to generalize the results to other areas with different living environments. Actually, the nationwide internet survey showed that awareness varies greatly by prefecture. The top three were Saga, with an awareness proportion of 76.9%, Miyazaki at 64.0%, and Tokushima at 58.5%; the lowest was Shizuoka, with a proportion of 34.6% [23]. It is difficult to say whether this discrepancy between prefectures resulted from selection bias (the top three prefectures all have small populations, so only a small number of people were surveyed from each one), or from differences in the degree of effort made by local municipalities to increase awareness. Thirdly, since the information was collected via questionnaire, there would be information biases such as recall bias. There could also be mistakes by participants while answering the questionnaire. Lastly, as mentioned earlier, since this is a cross-sectional study, different study is needed to elucidate the direction of causal relationship.

## 5. Conclusions

An association between social factors, especially attendance at regional events, and awareness of locomotive syndrome was revealed. In addition to strengthen awareness rising campaigns targeting men and for younger people, providing health education at social settings such as regional events may help improve future musculoskeletal health in the elderly.

## Figures and Tables

**Figure 1 ijerph-17-07272-f001:**
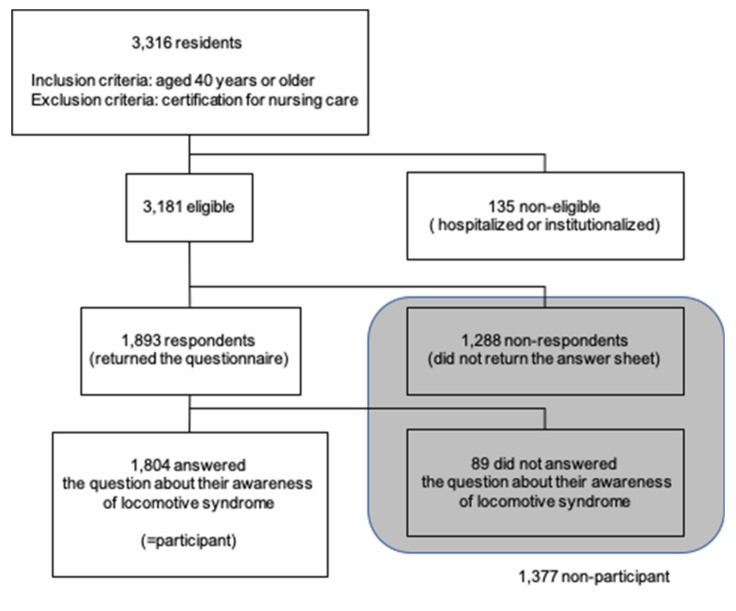
Schema of the recruitment of the participants.

**Table 1 ijerph-17-07272-t001:** Participant characteristics by sex, in association with the awareness of locomotive syndrome.

	Awareness of Locomotive Syndrome						
Characteristics	Men				Women		
	Yes/Total (*n*) ^a^	(%)	*p*-Value		Yes/Total (*n*) ^a^	(%)	*p*-Value
All							
	257/799	(32.2)			547/1005	(54.4)	
Age category							
40–49	25/92	(27.2)			61/121	(50.4)	
50–59	42/149	(28.2)			94/185	(50.8)	
60–69	89/253	(35.2)			176/268	(65.7)	
70–79	67/172	(39.0)			152/241	(63.1)	
80 or over	34/133	(25.6)	0.050		64/190	(33.7)	<0.001
Merital status							
Single/Divorced/Widowed	44/147	(29.9)			143/294	(48.6)	
Married	208/639	(32.6)	0.540		396/686	(57.7)	0.009
Educational attainment							
Junior high or lower	49/187	(26.2)			105/278	(37.8)	
High school or higher	205/600	(34.2)	0.042		433/705	(61.4)	<0.001
Experience of health promotion volunteer							
None			―		181/408	(44.4)	
Yes			―		360/573	(62.8)	<0.001

**^a^** Due to some missing values, the totals for the stratified subgroups are not equal.

**Table 2 ijerph-17-07272-t002:** Adjusted prevalence ratios of each covariates to increase awareness of locomotive syndrome in women.

		Awareness of Locomotive Syndrome		Model 1 ^b^ PR ^†^	95% CI ^⁑^	Model 2 ^c^ PR ^†^	95% CI ^⁑^
		Yes/Total (*n*) ^a^	(%)				
Basic characteristics							
Age group	40–49	61/121	(50.4)	Ref		Ref	
	50–59	94/185	(50.8)	1.01	0.81–1.27	0.92	0.73–1.15
	60–69	176/268	(65.7)	1.34	1.10–1.63	1.04	0.85–1.28
	70–79	152/241	(63.1)	1.47	1.20–1.80	1.11	0.90–1.38
	80 or over	64/190	(33.7)	0.83	0.62–1.10	0.69	0.51–0.94
Marital status	Single/Divorced/ Widowed	143/294	(48.6)	Ref		Ref	
	Married	396/686	(57.7)	1.02	0.90–1.17	0.99	0.86–1.13
Educational attainment	Junior high or lower	105/278	(37.8)	Ref		Ref	
	High school or higher	433/705	(61.4)	1.53	1.29–1.81	1.30	1.09–1.54
Lifestyle							
Alcohol drinking habit	No	353/677	(52.1)	Ref		Ref	
	Yes	190/322	(59.0)	1.06	0.94–1.19	1.02	0.91–1.14
Smoking habit	No	525/951	(55.2)	Ref		Ref	
	Yes	17/43	(39.5)	0.69	0.46–1.02	0.69	0.44–1.07
Weekly exercise habit	Less than one hour	301/683	(51.6)	Ref		Ref	
	More than one hour	217/341	(63.6)	1.16	1.04–1.30	1.05	0.94–1.18
Social Environment							
Participation to group activities	None	125/313	(39.9)	Ref		Ref	
	Yes	408/657	(62.1)	1.35	1.16-1.58	1.13	0.95–1.34
Attendance to regional events within one year	None/Did not know	218/490	(44.5)	Ref		Ref	
	Yes	320/499	(64.1)	1.34	1.19–1.50	1.26	1.10–1.43
Social support	None	4/9	(44.4)	Ref		Ref	
	Yes	538/983	(54.7)	0.94	0.41–2.14	0.68	0.32–1.44
Closeness to neighbors	None or greetings only	90/228	(39.5)	Ref		Ref	
	Daily talk or more	452/765	(59.1)	1.38	1.16–1.64	1.20	1.01–1.42
Experience of health promotion volunteer	None	181/408	(44.4)	Ref		Ref	
	Yes	360,573	(62.8)	1.31	1.15–1.49	1.25	1.09–1.43

**^a^** Due to some missing values, the totals for the stratified subgroups are not equal.**^b^** Adjusted by age group, marital status, and educational attainment. **^c^** Adjusted by all the variables shown in this table. Abbreviations: ^†^ PR prevalence ratio, ^⁑^ CI confidence interval.

**Table 3 ijerph-17-07272-t003:** Adjusted prevalence ratios of each covariates to increase awareness of locomotive syndrome in men.

		Awareness of Locomotive Syndrome		Model 1 ^b^ PR ^†^	95% CI ^⁑^	Model 2 ^c^ PR ^†^	95% CI ^⁑^
		Yes/Total (*n*) ^a^	%				
Basic characteristics							
Age group	40–49	25/92	(27.2)	Ref		Ref	
	50–59	42/149	(28.2)	0.98	0.64–1.50	1.00	0.66–1.53
	60–69	89/253	(35.2)	1.33	0.92–1.94	1.34	0.93–1.95
	70–79	67/172	(39.0)	1.52	1.03–2.24	1.54	1.04–2.30
	80 or over	34/133	(25.6)	1.11	0.70–1.74	1.20	0.74–1.94
Marital status	Single/Divorced/Widowed	44/147	(29.9)	Ref		Ref	
	Married	208/639	(32.6)	1.00	0.76–1.32	0.93	0.70–1.23
Educational attainment	Junior high or lower	49/187	(26.2)	Ref		Ref	
	High school or higher	205/600	(34.2)	1.38	1.04–1.83	1.28	0.96–1.70
Lifestyle							
Alcohol drinking habit	No	80/258	(31.0)	Ref		Ref	
	Yes	175/538	(32.5)	1.04	0.83–1.30	1.04	0.83–1.30
Smoking habit	No	199/617	(32.3)	Ref		Ref	
	Yes	57/181	(31.5)	0.99	0.77–1.28	1.01	0.79–1.30
Weekly exercise habit	Less than one hour	140/439	(31.9)	Ref		Ref	
	More than one hour	110/315	(34.9)	1.03	0.83–1.27	0.93	0.76–1.15
Social Environment							
Participation to group activities	None	60/223	(26.9)	Ref		Ref	
	Yes	195/563	(34.6)	1.23	0.95–1.59	1.10	0.84–1.44
Attendance to regional events within one year	None / Did not know	154/549	(28.1)	Ref		Ref	
	Yes	102/237	(43.0)	1.46	1.19–1.80	1.48	1.19–1.83
Social support	None	7/29	(24.1)	Ref		Ref	
	Yes	250/768	(32.6)	1.46	0.70–3.04	1.29	0.64–2.61
Closeness to neighbors	None or greetings only	59/217	(27.2)	Ref		Ref	
	Daily talk or more	198/580	(34.1)	1.15	0.88–1.50	0.97	0.74–1.28

**^a^** Due to some missing values, the totals for the stratified subgroups are not equal.**^b^** Adjusted by age group, marital status, and educational attainment.**^c^** Adjusted by all the variables shown in this table. Abbreviations: ^†^ PR prevalence ratio, ^⁑^ CI confidence interval.

## Data Availability

The datasets generated and analyzed during the current study are not publicly available because we did not receive consents for data provision to the third party.

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
