# Peer review of "Awareness of Locomotive Syndrome and Factors Associated with Awareness: A Community-Based Cross-Sectional Study"

_ijerph, 2020, doi:10.3390/ijerph17197272_

Round 1

Reviewer 1 Report

The authors have improved the manuscript

Reviewer 2 Report

The authors have submitted a much improved manuscript compared to the previous version.

Reviewer 3 Report

The paper has much improved, and although I have reservations about the interpretation of the data, and the strength of evidence for the clinical message, I think the article presents the data well enough for readers to judge themselves. 

This manuscript is a resubmission of an earlier submission. The following is a list of the peer review reports and author responses from that submission.

Round 1

Reviewer 1 Report

This work is focussed in a topic that is of increasing importance. There is an important increase of elderly people with a mobility limitation, that if addressed could be improved. Therefore, works like this one that gives information about them, are a great base as a beginning for planning actions.

  • Table 3. The format should be improved, as many of the figures are cut in two lines and it makes difficult to read the results.
  • Even though the main objective of this work is to check the degree of awareness, I think that in order to have a more complete picture of the problem, the authors should give some indications of what should be the interventions in order to improve the awareness and also in what actions are they thinking that would be desirable in order to improve the mobility in people.
  • I think that it is important that the interventions to improve mobility should be done in earlier as in very old people thus could be not possible any more or the benefits very few.

Reviewer 2 Report

The authors have submitted an interesting manuscript on locomotive syndrome awareness and factors associated with awareness. The work presents some points that the reviewer considers that they should be corrected before its possible publication. Abstract section The reviewer suggests that the authors define the acronym 95% CI. Methods section The reviewer advises the authors to describe the calculation of the sample size and study power. 2.2. Awareness of "lobomotive" syndrome. The reviewer harasses the authors to correct the typo: locomotive instead of lobotomive. 2.4. Statistical analysis Page 3, line 137, the reviewer advises the authors to properly define the abbreviation of "95% confidence interval" 95% CI instead of CI. Discussion section The reviewer advises the authors to argue why they conduct a study so similar to the "nationwide internet survey carried out in 2020 [24]". The authors could specify what are the advantages of their study compared to another so recent. In the limitations section, the reviewer advises the authors to include all study biases.

Reviewer 3 Report

The study titled Awareness of locomotive syndrome and factors associated with awareness: a community-based cross-sectional study, this a observational study was performed inn Koumi, Town (Japan).. The main weakness of the study regards the novelty, I'm not sure what is being added to the literature.

There are already studies showing the relation between knowledge about this questions as (to name ones):

Association between musculoskeletal function deterioration and locomotive syndrome in the general elderly population: A Japanese cohort survey randomly sampled from a basic resident registry https://www.researchsquare.com/article/rs-17805/v1

The Adverse Relationship of Locomotive Syndrome
https://www.jstage.jst.go.jp/article/jshhe/82/5/82_171/_pdf

ASSESSMENT OF LOCOMOTIVE SYNDROME: A PROSPECTIVE, OBSERVATIONAL STUDY IN COMMUNITIES OF GANDHINAGAR AND AHMEDABAD https://storage.googleapis.com/journal-uploads/wjpps/article_issue/1512469734.pdf

Other issues:

Material and Methods: In this section, you need to clearly describe how individuals were approached, how many were approached, how many were eligible, consented or refused. Also, Inclusion and exclusion criteria should be cited with references and cited guidelines for cross sectional observational study may be recommended in order to improve the quality of the manuscript.

The Discussion section is a rehashing of the results. It does not appear that the authors include much interpretation of what the study findings mean for clinical practice or research.

FInally, the conclusión is weak and too long.

We wish you all the best.